# Intraspecific Hybrids Versus Purebred: A Study of Hatchery-Reared Populations of Sterlet *Acipenser ruthenus*

**DOI:** 10.3390/ani10071149

**Published:** 2020-07-07

**Authors:** Sahana Shivaramu, Ievgen Lebeda, Vojtěch Kašpar, Martin Flajšhans

**Affiliations:** South Bohemian Research Center of Aquaculture and Biodiversity of Hydrocenoses, Faculty of Fisheries and Protection of Waters, University of South Bohemia in České Budějovice, Zátiší 728/II, 389 25 Vodňany, Czech Republic; ilebeda@frov.jcu.cz (I.L.); vkaspar@frov.jcu.cz (V.K.); flajsh@frov.jcu.cz (M.F.)

**Keywords:** hybrid, sturgeon, aquaculture, restocking, fitness-related traits

## Abstract

**Simple Summary:**

Sturgeon are an ancient fish family widely distributed in the northern hemisphere, and many are listed as critically endangered, facing risk of extinction in the wild due to habitat destruction and overfishing to supply caviar. As well as breeding for commercial aquaculture, conservation programs use stock derived from other areas along with local native restocking projects. The fitness of introduced non-native sturgeon and their progeny from crossing with native stock is unknown. To assess the consequences of such interbreeding populations that have adapted to local conditions, we first analyzed the genetic makeup of hatchery-reared Danube and Volga populations of sterlet *Acipenser ruthenus* to ensure that the two were distinguishable. We then investigated the fitness-related traits and genetic diversity of the purebred fish and their hybrid crosses. The hybrids grew faster and exhibited a higher genetic diversity than purebred fish, suggesting their superior fitness for commercial aquaculture. We did not observe adverse effects of the population cross, and an investigation of future generations may provide insight into the consequences of using non-native stock in conservation programs.

**Abstract:**

Hatchery-reared sterlet originating from the Danube and Volga river basins that showed population-discriminatory alleles on at least one microsatellite locus were used to produce purebred (within-population) and hybrid crosses to evaluate intraspecific hybridization with respect to the genetic polymorphism and physiological fitness of fish for commercial aquaculture and, conservation programs. Reciprocal crossing assessed the effect of parent position. The fish were reared in indoor and outdoor tanks and monitored over 504 days for growth traits. The highest final mean body weight (144.9 ± 59.5 g) was recorded in the Danube (♀) × Volga (♂) hybrid and the highest survival in the Volga (♀) × Danube (♂) hybrid. The Volga purebred exhibited the lowest mean body weight (124.8 ± 57.6 g). A set of six microsatellites was used to evaluate the heterozygosity. The mean number of alleles was highest in the Danube (♀) × Volga (♂) hybrid and lowest in the Volga purebred, suggesting an influence of the parent position in the hybridization matrix. The higher level of genetic polymorphism, as in the Danube (♀) × Volga (♂) hybrid, may confer greater fitness in a novel environment. Our analysis revealed that the intraspecific hybrids performed better than the purebred fish in the controlled and suboptimal rearing conditions.

## 1. Introduction

Acipenseriformes are one of the oldest groups of fish, having evolved for more than 200 mya, and are extensively distributed in the northern hemisphere [1]. Sturgeons (family Acipenseridae) have been classified as critically endangered fishes on the IUCN (International Union for Conservation of Nature) Red List [2], and the populations of these species have drastically decreased mostly due to overfishing, the destruction of their natural habitat, and water pollution. The overexploitation of wild populations has led to the listing of all sturgeon species in the Appendices to CITES (Convention on International Trade in Endangered Species of Wild Fauna and Flora) over the past four decades for restocking and caviar production [2]. Besides this, sturgeons are artificially propagated in aquaculture, mainly for the production of black caviar and boneless meat. Around 102,327 t of sturgeon meat and 364 t of black caviar is estimated to have been produced in 2017 [3]. The most commercially important sturgeon species reared for caviar and meat production are *A. baerii, A. gueldenstaedtii*, and the hybrid *H. dauricus × A. schrenckii* [3].

The sterlet *Acipenser ruthenus* is one of several sturgeon species found in Eurasian waters, inhabiting rivers flowing into the Caspian, Black, Azov, Baltic, White, Barents, and Kara Seas [4]. Compared to other sturgeon species, it is relatively small and has a shorter reproduction cycle. The onset of sexual maturity occurs from 3 to 7 years in males and 5 to 12 years in females [5,6]. Although the commercial importance of *A. ruthenus* in aquaculture is low, the species is often used in hybridization programs—for example, in the production of the bester, a hybrid of the great sturgeon (beluga) *Huso huso* female and a sterlet male [7]. It is listed as threatened on the IUCN Red List [2].

Overfishing, poaching, habitat destruction, and other anthropogenic causes have led to the profound decline of sterlet in the Upper and Middle Danube River from the years 1996 to 2010 [8,9]. The annual catches of sterlet in the Danube river varied between 36 and 117 metric tons from 1958 to 1981, and the construction of the Derdap dam in 1970 led to a more than 50% decrease in sterlet catches [10]. Restocking is being undertaken in the Danube using remaining specimens as broodstock along with non-native stock from neighboring river systems in Romania and Hungary [11,12,13]. Projects such as STURGENE conduct targeted restocking in the lower Danube [14]. Recently, “LIFE Sterlet: Restoration of sterlet populations in the Austrian Danube” has begun restocking of sterlet in the Upper Danube [15]. Reinartz et al. [16] revealed that the wild population of Danube sterlet currently includes about 23% non-native sterlet carrying a partial or complete Volga genotype, pointing to a need for the genetic analysis of fish used for restocking.

Genetic rescue is nowadays considered as an effective conservation measure to address genetic destruction associated with habitat loss and fragmentation [17]. The genetic variability in wild populations of fish at risk of extinction is considered a key element of their adaptability to extreme climate change and anthropogenic pressure and, hence, may play a crucial role in stocking success [15,18,19]. However, we should note that the crossing of two genetically distinct individuals can both reduce the probability of extinction and dramatically increase fitness, which can lead to elevated population size. The major river catchments occupied by large historical populations of European sturgeon flow to either the northeast Atlantic Ocean or to one of the surrounding seas (Baltic, Black, Mediterranean, North). Many locally adapted sturgeon populations have become extinct or decreased to the point that genetic heterogeneity is extremely limited [20]. The question of the optimal source of recovery stock is critical to the initial phase of recovery programs [21], and the investigation of genetic patterns within and among populations is a prerequisite for a successful restocking program [15,16]. A potential option is to expand genetic heterogeneity through intraspecific hybridization. Recently, a genetic rescue study on *Poecilia reticulata* concluded that the gene flow between small populations may be essential for providing the necessary variation for populations to persist and adapt to fast-paced environmental change [22]. Unlike interspecific hybridization, within-species crosses usually produce viable and fertile individuals; however, the fitness of offspring can be higher (heterosis) or lower (outbreeding depression) than that of their parent populations.

The production of intraspecific hybrids of different populations is also an efficient strategy in aquaculture. The most widely recognized short-term benefit of intraspecific crossing is hybrid vigor, or heterosis, which is exhibited in the superiority of phenotypic traits of the hybrids over those of their parent stock [23,24,25]. The phenomenon can exert impacts on fitness traits that are independent of adaption to a local habitat as well as to those that are environment-specific. A feasible option for conservation stocking may be the mixing of fish from isolated populations to increase genetic diversity, allowing future selection to play out under local natural pressures [15].

Heterosis is not considered detrimental to conservation efforts, especially when it is undertaken using different populations within species with same chromosome numbers, which leads to the production of fertile offspring. The primary aim of selecting potential populations should be to avoid the outbreeding depression that may result from cross-breeding populations with fixed chromosome differences originating from different environments for over 20 generations or with no gene flow for over 500 years [26,27]. To avoid outbreeding depression when more than one population is used for restocking, it is best to choose parent populations that show low genetic divergence and are derived from similar local habitats [27,28,29].

There are currently no reports of the evaluation of introgression asymmetry through hybridization in sturgeon associated with the extinction risk and fitness of future back-crossed generations. It is essential to expand the knowledge of genetic characteristics and fitness fate of hybrids produced in natural ecosystems. There have also been no studies documenting the implications of the interbreeding of divergent sterlet populations in the controlled environment of a hatchery.

We evaluated the effects of intraspecific hybridization at the species level with respect to the fitness-related traits such as body weight and specific growth rate, along with growth heterogeneity and cumulative survival at the population level within a closed aquaculture system provided with common environment, using sterlet as a model species. In addition, the genetic polymorphism and heterozygosity were assessed in purebred fish and intraspecific hybrids using six microsatellite markers.

## 2. Materials and Methods

This experiment was conducted at the Genetic Fisheries Center, Faculty of Fisheries and Protection of Waters (FFPW) in Vodňany, Czech Republic.

### 2.1. Ethical Statement

The experiment was carried out in compliance with the criteria of the Animal Research Committee of the FFPW. The fish were maintained in accordance with the principles of the animal welfare act of the Czech Republic and laboratory animal care in compliance with the law on the protection of animals against cruelty (Act no. 246/1992 Coll., ref. number 16OZ15759/2013-17214).

### 2.2. Parental Populations

We used adult sterlet broodstock from separate captive-bred, hatchery-reared populations originating from the Danube and Volga Rivers to produce hybrid and purebred offspring. The Volga broodstock was obtained from the Genetic Fisheries Center, FFPW, Vodňany, and the Danube broodstock from the Velký Dvůr fish hatchery operated by Rybnikářstvi, Pohořelice a.s., Czech Republic. The genetic origin of the Danube broodstock was the Danube River at the territory of the Slovak Republic, and the stock had been bred in captivity for a minimum of three generations. The Volga population was descended from specimens imported from Russia and bred in captivity for at least three generations. The Volga and Danube rivers are the two largest river systems in Europe with varying water flow regimes in their ranges. The Danube River is located in central and eastern Europe, flowing east from the north slopes of the Alps to the western Black Sea. The Volga River is located in central Russia and flows south from a broad interior region to the northern Caspian Sea. Reinartz et al. [16] have already demonstrated the genetic differences among the sterlet stocks originating from the Danube and Volga rivers based on microsatellite markers.

### 2.3. Sampling for the Assessment of Population Divergence

We collected fin clips from 100 brooders of each population and stored them in 96% ethanol for use in microsatellite genotyping. Twelve males and 12 females from any population that expressed discriminatory alleles in at least one microsatellite loci were used as the broodstock to produce six purebred and six hybrid groups of Danube and Volga sterlet.

### 2.4. Broodstock Handling and Hormone Induction

The broodstock were held in the controlled conditions of a re-circulating water system of 5 m^3^ indoor tanks maintained at 15 °C for a seven-day acclimation period prior to hormone stimulation. The fish were immersed in 0.07 mL L^−1^ of clove oil for anesthesia before handling.

Spermiation was induced by an intramuscular injection of 4 mg kg^−1^ body weight (BW) carp pituitary suspension in physiological saline 36 h before the intended sperm sampling [30]. Ovulation was stimulated with an initial injection of 0.5 mg kg^−1^ BW carp pituitary suspension in physiological saline 42 h before the intended ovulation, followed by a second injection after 12 h of 4.5 mg kg^−1^ BW with the same suspension [30]. The eggs were collected via the micro-incision of oviducts following the protocol described by Štěch et al. [31] and maintained in aerobic conditions at temperatures below 16 °C during the evaluation of gamete parameters, such as spermatozoon motility and egg quality, prior to fertilization.

### 2.5. Fertilization and Hatching

To establish hybrid and purebred crosses, a coordinated simultaneous breeding program was organized at Vodňany and Pohořelice. Twelve males and 12 females from each population were used to produce two purebred and reciprocal hybrid crosses. Three sub crosses were established in a main cross. Each sub cross was established using four females and four males out of 12 females and 12 males of the respective population. The breeding program is shown in Figure 1. Milt from the Volga males was held separately at 4 °C and transported from Vodňany to fertilize the Danube eggs in the fish hatchery at Pohořelice. Three Danube (♀) × Danube (♂) and three Danube (♀) × Volga (♂) crosses were produced and incubated at Pohořelice. Likewise, the milt from individual males of the Danube population was separately held at 4 °C and transported to fertilize Volga eggs at the Genetic Fisheries Center, Vodňany. Three Volga (♀) × Volga (♂) and three Volga (♀) × Danube (♂) crosses were produced and incubated at Vodňany. The fertilized eggs from the six established crosses at Pohořelice were transferred at the neurula stage to Vodňany in oxygenated water at 15 °C. The hatching and initial rearing was conducted in Vodňany.

For fertilization, 50 g of eggs were collected from each of four females. The eggs were pooled, divided into four plastic beakers in 50 g aliquots, and placed on an electronic shaker at 200 rpm and 10 mm deflection. The aliquots were separately fertilized by individual males to avoid sperm competition and to balance the genetic contribution of individual males. A clay suspension (20 g L^−1^) was subsequently added to eliminate egg stickiness [24], and, after shaking for an additional 45 min, the eggs were repeatedly washed in water, pooled, and incubated in glass jar incubators in triplicate. Incubators were supplied with UV-sterilized re-circulating tap water at 15.0 °C, 9 mg L^−1^ O_2_. For estimating the fertilization rate, ~100 eggs from each incubator were randomly sampled at 6 h post-fertilization in triplicate, and the live embryos were counted at the 2nd or 3rd cleavage division [32]. The sub crosses were pooled, resulting in the establishment of two purebred groups (Danube and Volga) and two reciprocal hybrid groups (D × V and V × D). Forty-five swim-up larvae from each group were collected and preserved in 96% ethanol for molecular analysis. Larval rearing was conducted at the Genetic Fisheries Center in Vodňany.

### 2.6. Rearing of Progeny Groups

The fish were reared according to standard aquaculture procedures in the indoor recirculating aquaculture system and outdoor tanks. The fish groups were reared in three tank replicates throughout the experiment, be it indoor or outdoor tanks, or separate or communal rearing. After the complete absorption of the yolk sac, the larvae were fed with diced sludge worms *Tubifex tubifex* for two weeks ad libitum and then shifted to co-feeding with dry feed. After four weeks of co-feeding, the larvae were completely transferred to dry feed (Alltech Coppens, Helmond, The Netherlands) and fed daily at a feeding rate of 8% of the fish biomass for the first three months (Table 1). Approximately 30% mortality was recorded during the shift from live to dry feed. During the first three months, the stock in 0.5 m^3^ recirculating indoor tanks was regularly reduced in the following manner to accommodate growth and limit the density during the critical growth period. After the initial three months of rearing in indoor troughs, the fish were transferred to 3.2 m^3^ separate indoor circular tanks at an initial density of ~7 kg m^−3^ and fed daily at a feeding rate of 4% of the fish biomass. At 175 days post-hatching (dph), the surviving fish were implanted with subcutaneous individual passive integrated transponder tags (134.2 kHz; AEG, Germany). At 229 dph, they were moved to 3.2 m^3^ outdoor circular tanks for communal rearing in a density of ~10 kg m^−3^ and fed daily at a feeding rate of 4% of the fish biomass. The water-dissolved oxygen and temperature were kept optimum throughout the rearing period. The feed pellet size was adjusted according to the fish developmental stage (Table 1).

### 2.7. Measurement of Performance and Mean Heterosis

The fish were measured to calculate the mean weight and cumulative survival at 77, 175, 229, 325, 386, and 504 dph. The specific growth rate (SGR, % day^−1^) was calculated as SGR = (lnW_f_ – lnW_i_ × 100)/t, in which W_i_ and W_f_ are the initial and final mean BW and t is the time interval between the samplings in days. The growth heterogeneity (GH) was calculated from CV_FBW_/CV_IBW_, where CV is the coefficient of variation (100 × SD/mean) and IBW and FBW are the initial and final mean BW. The average heterosis for the mean BW and cumulative survival rate of the hybrid fish were calculated as average heterosis = [(F1 − MP)/MP] × 100, in which F1 = the value of hybrid, MP = the mean value of the purebred crosses.

### 2.8. Microsatellite Marker Analysis

Whole genomic DNA was extracted using the Nucleo Spin^®^ Tissue kit (MACHEREY-NAGEL GmbH and Co. KG, Düren, Germany) from fin clips for the population divergence study and from swim-up larvae for the assessment of heterozygosity in groups. Six microsatellite markers, AciG 35 [33], AfuG 135 [34], Aox 45 [35], Spl 101, Spl 163, and Spl 173 [36] were used for PCR amplification carried out according to Havelka et al. [37]. A microsatellite fragment analysis was conducted on a 3500 ABI Genetic Analyzer (Applied Biosystems, Waltham, USA) using the GeneScan LIZ 600 size standard (Applied Biosystems), and genotypes were identified in the Genemapper 4.1 software (Applied Biosystems, Waltham, USA). The mean number of effective alleles (N_A_), pairwise G_ST_, D_A_ matrix, gene diversity at each locus, fixation index, and expected (H_e_) and observed (H_o_) heterozygosity of the Danube and Volga populations were calculated using GeneAlex [38]. Likewise, the mean number of alleles (N_A_) and expected (H_e_) and observed (H_o_) heterozygosity used to assess the level of polymorphism in the analyzed progeny fish groups were calculated using GeneAlex [38]. The visualization of the genetic relationships among the progeny fish groups and two populations based on the multilocus genotypes were performed by a factorial correspondence analysis (FCA) in the GENETIX software (Version 4.05, 2004) for MS Windows. This enabled the visualization of the data in multidimensional space with no a priori assumptions on grouping, using each allele as an independent variable.

### 2.9. Statistical Analysis

A statistical analysis was conducted using Statistica 13 (STATISTICA advanced module STATISTICA Multivariate Exploratory Technique; Statsoft). The data were analyzed for normal distribution using the Kolmogorov–Smirnov test. Multiple comparisons were carried out by a one-way analysis of variance (ANOVA) at a significance of *p* = 0.05, and Tukey’s post-hoc (parametric data), Kruskal–Wallis, and Dunn’s post-hoc tests were used for the non-parametric data fertilization rate, the hatching rate, the mean BW, the growth heterogeneity, and the specific growth rate. The differences in survival between the three tank replicates were evaluated using a Pearson’s Chi-square test at a significance of *p* = 0.05. The significance of the differences between the individual groups was tested by an ANOVA with a significance at *p* = 0.05. The significance of the differences in N_A_, H_o_, and H_e_ between the Danube and Volga populations and between the hybrids and purebreds was tested with a one-way ANOVA at a significance of *p* = 0.05.

## 3. Results

### 3.1. Population Genetic Analysis of Danube and Volga Sterlet

The highest allele diversity was found in the Danube population for locus Aox 45 (0.882) (Table 2). The allele diversity among the loci was higher in the Danube population than in the Volga population. The observed heterozygosity and expected heterozygosity were highest in the Danube population (Table 3). The pairwise G_ST_ and D_A_ matrix values of the Danube and Volga populations were 0.136 and 0.549, respectively. These values indicate a moderate genetic differentiation between the Danube and Volga stock.

The fixation index values of the Danube and Volga populations are shown in Table 3. The factorial correspondence analysis (FCA) grouped the populations into two distinct clusters with no overlap (Figure 2).

### 3.2. Performance Comparison of Purebreds and Hybrids

The highest fertilization rate (81.73 ± 2.16) was recorded in the D × V hybrid, and the highest hatching rate was in the Danube purebred (75.85 ± 1.89) (Figure 3). The lowest fertilization (74.69 ± 3.35) and hatching rates (41.10 ± 8.3) were observed in the purebred Volga group. The fertilization rate differed significantly between the D × V hybrid and Volga purebred, and the hatching rate significantly differed between the purebred Danube and Volga groups. Although the fertilization rate was highest in the D × V hybrid, the hatching rate was significantly lower than in the Danube purebreds.

The observed mean BW and cumulative survival at different sampling points are shown in Table 4. At the conclusion of the study (504 dph), the highest mean BW (144.98 ± 59.51) was recorded in the D × V hybrid, and highest survival was observed in the V × D hybrid (Figure 4). The lowest BW at 504 dph (124.8 ± 57.6) was recorded in the Volga purebred. The lowest cumulative survival was found in the D × V hybrid during all the sampling points. The average heterosis for BW was highest, and positive, in the D × V hybrid at most sampling points (13.40% at 504 dph), but this group displayed negative values of average heterosis with respect to the cumulative survival at all the sampling points.

The SGR was significantly higher in the D × V hybrid (3.10 ± 0.02) and significantly lower in the V × D hybrid (1.77 ± 0.1) from 58–175 dph, compared to other groups. There were no significant between-group differences in the SGR at later sampling points. Growth heterogeneity differed significantly among the groups from 58 to 175 dph, with the D × V hybrid exhibiting the highest GH and the V × D hybrid the lowest. There were no significant GH differences between the Danube and Volga purebreds at the later sampling points, whereas the reciprocal hybrids showed significant differences in the GH from 386–504 dph (Table 5).

### 3.3. Microsatellite Marker Analysis

The mean number of alleles was significantly higher in the D × V hybrids, and the Volga purebred displayed the lowest mean number of alleles (Table 6). The observed (0.6962 ± 0.0498) and expected (0.7589 ± 0.0685) heterozygosity were highest in the D × V hybrid (Table 6). The genotype data obtained from six microsatellite loci were submitted to an FCA, which grouped the Danube and Volga purebreds into two distinct clusters without overlap. The D × V and V × D clusters occupied positions intermediate between the Danube and Volga purebred clusters, with little overlap between them (Figure 5).

## 4. Discussion

### 4.1. Genetic Analysis of Hatchery-Reared Populations of Volga and Danube Sterlet

At first, the genetic structure of the two geographically distant populations—i.e., Volga sterlet and Danube starlet—was investigated in this study to ensure that the sample sets of the populations displayed genetic heterogeneity, since it is reported that 23% of wild Danube sterlet possess a partial or complete Volga genotype [16]. The Danube and Volga sterlet were found to be genetically distinct. The genetic divergence between the Danube and Volga stock was ideal for the study, but the Volga sterlet showed a lower intra-population genetic variation than was observed in the Danube population. Although three microsatellite loci revealed a considerable degree of intra-population genetic variation in the Volga sterlet, the overall genetic variation in the Volga sterlet was lower than in the Danube sterlet. A combination of intrinsic and extrinsic parameters may have influenced the genetic variation in the studied hatchery-reared stock. Reproductive factors (small number of broodstock, sex ratio biases, inbreeding, high individual variation in fecundity) and fluctuations in the population size over a period of time can lead to the loss of allelic diversity in fish populations [6]. Although accurate ecological data pertaining to native Danube and Volga sterlet stock were not available, the genetic and phenotypic differences observed provide information on the effect of hybridization and its potential importance in aquaculture and consequences in the wild.

Regular monitoring of the genetic diversity of broodstock and progeny is crucial to maintaining the genetic integrity of a fish farm population [6]. The selection of individual broodstock characterized by high genetic variation is essential when the genetic diversity drops to a critical level [39].

### 4.2. Fitness-Related Traits of the Progeny Groups

We evaluated the relative fitness of the purebred and inter-population crosses, rearing hatchery-derived Danube and Volga sterlet under identical conditions. Significant between-group differences were detected in fitness-related traits, suggesting a variation in the potential for fish farming. Hybridization is commonly used in aquaculture to increase fitness, manifested as an improvement in relevant physiological traits compared to the parent species [40,41]. The reproductive characteristics of many interspecific sturgeon hybrids have been widely studied [41,42,43], but similar aspects of intraspecific hybrids are not well-documented. However, in aquaculture reproductive factors may be of secondary concern, with growth being the primary interest [40].

The hybrids showed a higher mean BW compared to either purebred at most sampling points. Our results are consistent with those reporting an effect of the parent position in the hybridization matrix in the Mekong giant catfish *Pangasianodon gigas,* rohu *Labeo rohita*, and common carp *Cyprinus carpio* [44,45,46]. The D × V hybrid displayed the highest mean BW at four of the six sampling points, possibly a maternal effect. Females may exert a stronger influence than males on the phenotypic expression of offspring traits [46]. The maternal effect may be due to the mother’s nuclear and extra-nuclear genes [47].

Our results are also in agreement with the results of studies exploring sturgeon hybridization to increase the growth rate and improve productivity through hybrid vigor [21,41,48,49,50]. We found a dramatic effect of hybridization on cumulative survival, both negative and positive. The observed values of cumulative survival in the V × D hybrids were highest at three of the six sampling points, whereas the D × V hybrids showed the lowest cumulative survival at five sampling points. The cumulative survival in the D × V hybrids decreased massively on the second sampling point itself (175 dph), which negatively influenced the cumulative survival of this group during the next sampling points. The mortality recorded in the D × V hybrids on 175 dph can be mainly due to cannibalism and the higher density. However, the recorded mortality could be associated with various intrinsic (genetic) and extrinsic (environmental) factors affecting the experimental groups throughout the experiment. Once the larvae switches to formulated feed, it can have a significant impact on the individual larvae’s growth and can lead to individual-specific differences in size. The pellet size of the feed fed to different developmental stages should be seriously considered, otherwise the individual-specific differences in size can result in cannibalism. Cannibalism can directly be associated with dramatic decline in the fish survival. Sometimes, a decreased growth rate can also leads to decreased survival via prolonged stage duration, which can also increase the predation risk and size-dependent mortality [51]. Memis et al. [42] observed that the survival rate declined to 27% at 75 dph in Russian sturgeon purebreds. Although the stocking density was the same for replicate tanks in the beginning, the cannibalism might have led to tank-specific differences in cumulative survival during the later sampling points. Interestingly, the observed values of cumulative survival in the V × D hybrids were located between those of the parent species at half of the sampling points. The source of a significant portion of phenotypic variation is likely genetic adaptation to the local environment [52,53], as influenced by such biotic factors as degree of competition, predation, and population density, along with abiotic factors including temperature, levels of dissolved oxygen, ammonia, photoperiod, and feed quality and availability. Therefore, an important conclusion that can be drawn from our study is that the consequences of hybridization are trait-specific. The lower survival rate recorded in the D × V hybrids may have resulted from outbreeding depression, a potential concern for conservation or reintroduction efforts involving hybrids of Volga and Danube populations.

### 4.3. Genetic Analysis of the Progeny Groups

Hybridization may lead to an increase in genetic polymorphism and heterozygosity [40], resulting in improved growth and other aspects of fitness [54,55]. The level of heterozygosity was found to be significantly higher in the D × V hybrid compared to the purebreds. No significant difference in genetic polymorphism was observed in the V × D hybrid compared to the purebreds. This may be a maternal effect, since we observed a lower genetic variation in the Volga sterlet broodstock compared to the Danube sterlet. The hybrids possessed a higher mean number of alleles compared to both purebreds, which might have implications for heterosis and adaptations providing higher fitness in a novel environment [56]. Therefore, the observed phenotypic divergence could be due to the differences in population genetic pools.

The competitive risk to wild populations from intraspecific hybrids is generally less than that from interspecific hybrids, since the level of genetic polymorphism and the occurrence of rare alleles is comparatively lower. Also, intraspecific hybrids possess the same number of chromosomes, eliminating the production of sterile hybrids. Nevertheless, intraspecific hybridization can present risks to future generations associated with outbreeding depression that should be considered in the selection of non-native broodstock for increasing productivity [29]. The escape of non-native fish from aquaculture facilities can pose a significant threat to the native stock genetic variation and give rise to genomic introgression. Crosses of native with non-native sterlet can reduce their habitat-specific adaptations and lead to the dilution or elimination of the selective advantages present in the native sterlet [16].

The sterlet hybrids produced in the present study can be exploited for commercial aquaculture to increase growth rate, improve productivity through hybrid vigor, and transfer desirable traits to future generations. They can be potential candidates for enclosed common-environment studies to determine their fate under target wild conditions. Researchers have studied the consequences of stocking multiple strains of Atlantic salmon *Salmo salar* and assessed the correlation between genetics and reproductive quality in their broodstock [27]. Although the results showed no signs of outbreeding depression in the F1 generation, research on the fate of F2 back-crosses and subsequent generations would be of value, because previous studies have suggested that outbreeding depression is an outcome of intrinsic factors (factors influenced by genotypes), which are most likely to manifest in the F2 generation or later, when the original parental genomes on the same chromosome are subjected to recombination [27].

Our analysis showed the intraspecific hybrids to perform better than the purebreds, similar to our previous results on interspecific sturgeon hybrids [41]. There were significant differences in the mean BW and cumulative survival of the reciprocal hybrids in this study. This suggests that the level of hybrid vigor may be related to the parent position in the hybridization matrix. There appeared to be no fitness-related disadvantages to the intraspecific hybridization of Volga and Danube sterlet in controlled and common-environment hatchery conditions, at least in the F1 generation studied.

In addition to genetic and phenotypic plasticity as a source of the observed differences in fitness-related traits, we cannot exclude likely environmental and sampling biases. The growth rates of the fish under study were lower than those commonly reported, which could be because of the suboptimal rearing environment. The fish with lost tags in the communal tanks were excluded from the analysis, which may have influenced the average measures of mean body weight and survival. The effect of cannibalism and tank-specific population density (in the earlier developmental stages) on the depressed growth rates and cumulative survival cannot be neglected. However, as these fish were reared in pooled stock, the examined fitness-related traits were equally affected by unfavorable rearing conditions. Hence, we believe that the observed differences were most likely caused by genetic origin (purebred vs. hybrid). To the best of our knowledge, this study brings the first observation of the effect of intraspecific hybridization on fitness-related traits in hatchery-reared populations of sterlet. Therefore, it has an important implication for additional comparative studies focusing on the survival and fitness-related traits of hybrid crosses in the natural environment. Nevertheless, any generalization of our results to conventional sturgeon aquaculture should be undertaken with caution.

## 5. Conclusions

The hatchery-reared Danube and Volga sterlet broodstock differed genetically. The D × V hybrid grew faster than the purebred sterlet, and its reciprocal hybrid (V × D) showed a higher survival at three of six sampling points compared to the purebred. The observed genetic diversity was significantly higher in the D × V. Our results indicate that the D × V hybrid shows potential for achieving a better growth rate and can be recommended over purebreds for commercial aquaculture. Future research focusing on the investigation of the fecundity, gamete quality, survival of F1 hybrids as well as the study of fitness-related traits in future generations should be undertaken in a more natural setting or, ideally, the native habitat. This can provide insight into the fate of intraspecific sturgeon hybrids, which could serve as potential information for conservation efforts.

## Figures and Tables

**Figure 1 animals-10-01149-f001:**
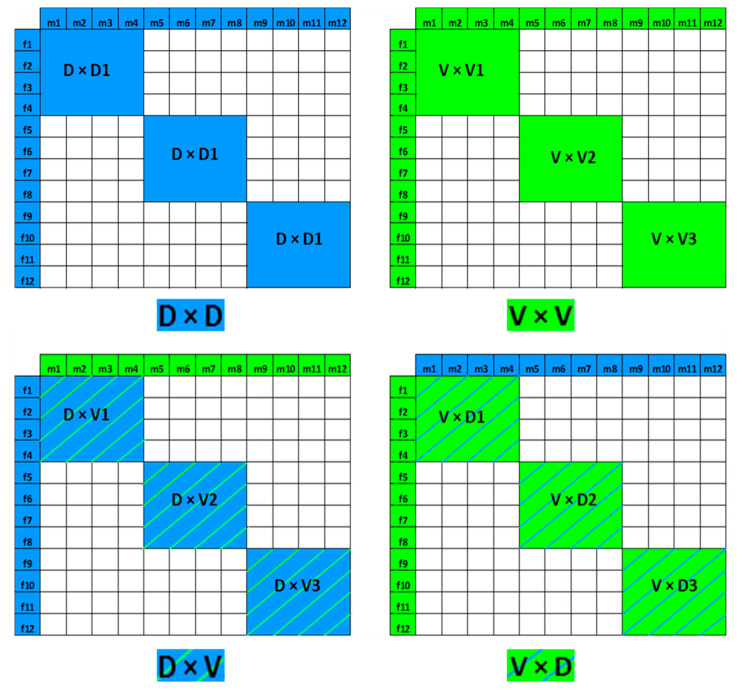
The cross-breeding of sterlet, *Acipenser ruthenus*, using 12 males and 12 females from Danube (D, blue) and Volga (V, green) populations, establishing three groups of purebred Danube and Volga (D1–D3, V1–V2) and six groups of reciprocal hybrids.

**Figure 2 animals-10-01149-f002:**
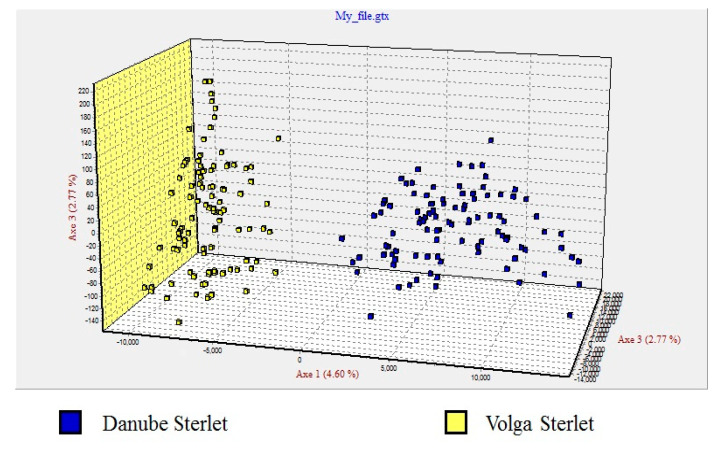
Factorial correspondence analysis based on six microsatellite loci illustrating genetic relationships among the Danube sterlet (blue squares) (*n* = 100) and Volga sterlet (yellow squares) (*n* = 100).

**Figure 3 animals-10-01149-f003:**
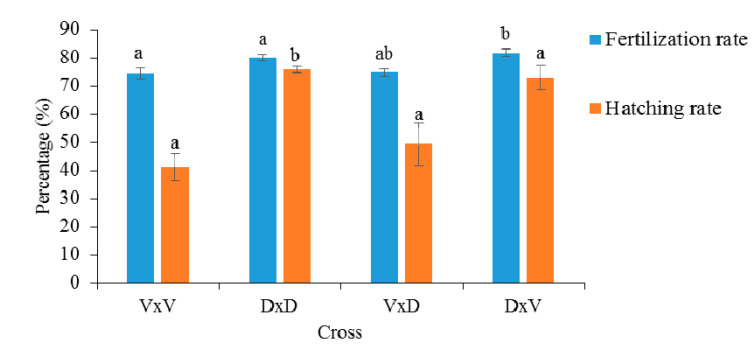
Fertilization and hatching rates of the analyzed hybrid and purebred Volga and Danube sterlet. Columns with the same superscript do not differ significantly at *p* < 0.05.

**Figure 4 animals-10-01149-f004:**
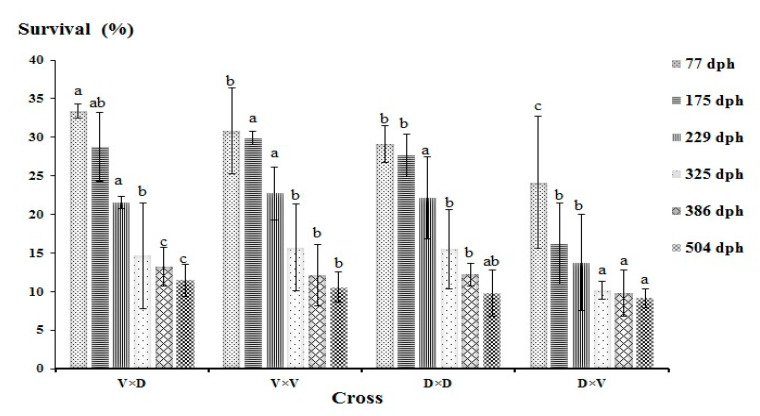
Observed values of cumulative survival (% mean ± SD) among the analyzed hybrid and purebred sterlet at selected days post-hatching (dph). Different superscripts within a row indicate significant differences.

**Figure 5 animals-10-01149-f005:**
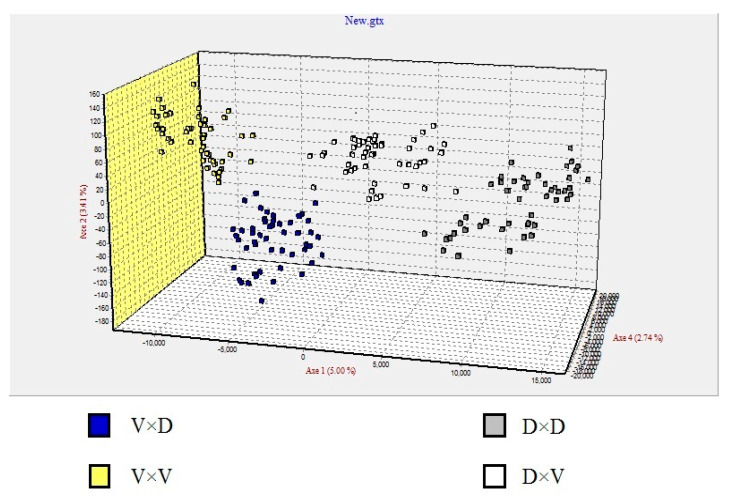
Factorial correspondence analysis based on six microsatellite loci illustrate the genetic relationships among the purebred and hybrid crosses. Volga sterlet purebred (yellow squares) (*n* = 48), Danube sterlet purebred (grey squares) (*n* = 48), Volga sterlet × Danube sterlet (blue squares) (*n* = 48), and Danube sterlet × Volga sterlet (white squares) (*n* = 48).

**Table 1 animals-10-01149-t001:** Proximate composition and pellet size of the feed relative to the fish developmental stage.

Fish Weight (g)	Rearing System	Days Post-Hatching	Life Stage	Feed	Feed Size (mm)	Protein (%)	Fat (%)	Crude Fiber (%)	Ash (%)	Total *p* (%)
0.2–0.5	Indoor troughs	1–11	Larvae	Alltech Coppens^®^ Advance	0.2–0.5	56	15	0.1	12	1.99
0.5–1.5	Indoor troughs	12–29	Fry	Alltech Coppens^®^ Advance	0.5–0.8	56	15	0.1	12.0	1.99
1.5–5.0	Indoor circular tanks	30–58	Early fingerling	Alltech Coppens^®^ Start Premium	1.0	54	15	0.3	10.3	1.73
5.0–10	Indoor circular tanks	59–85	Fingerling	Alltech Coppens^®^ Start Premium	1.0/1.5	54	15	0.3	10.3	1.73
10–50	Indoor circular tanks	86–228	Early juvenile	Alltech Coppens^®^ Alevin	2.0	54	15	1.1	9.0	1.32
50–100	Outdoor circular tanks	229–325	Early juvenile	Alltech Coppens^®^ Alevin	2.0	54	15	1.1	9.0	1.32
100–200	Outdoor circular tanks	326–504	Early Juvenile	Alltech Coppens^®^ Supreme-15	3.0	49	10	1.5	7.9	1.27

**Table 2 animals-10-01149-t002:** Allele diversity per locus and population.

Locus	Danube	Volga
Spl 163	0.811	0.772
Spl 101	0.743	0.831
Spl 173	0.589	0.650
AfuG 135	0.605	0.763
Aox 45	0.882	0.578
AciG 35	0.722	0.602

**Table 3 animals-10-01149-t003:** Summary statistics of the genetic variation in the Danube and Volga populations of sterlet.

Locus	H_o_	H_o_ SD	H_e_	H_e_ SD	N_A_	F
Danube	0.7346 *	0.0133	0.7231	0.0482	5.7 *	0.165
Volga	0.6862 *	0.0287	0.7018	0.0197	5.2 *	0.132

N_A_ = Mean no. of alleles; H_e_ = expected heterozygosity; H_o_ = observed heterozygosity; F = fixation index. * indicates significant difference at *p* < 0.05.

**Table 4 animals-10-01149-t004:** Observed values of body weight (g mean ± SD) and average heterosis (%) of hybrid and purebred sterlet at selected days post-hatching (dph). Different superscripts within a column indicate significant differences.

	58 dph (*n* = 240)	175 dph (*n* = 240)	229 dph (*n* = 240)	325 dph (*n* = 240)	386 dph (*n* = 240)	504 dph (*n* = 240)
Body weight (g)						
V × D	5.28 ± 2.38 ^b^	42.7 ± 20.50 ^a^	61.40 ± 26.11 ^a^	80.20 ± 36.62 ^a^	107.17 ± 36.97 ^ab^	137.73 ± 58.45 ^b^
V × V	5.09 ± 2.05 ^b^	43.21 ± 17.23 ^a^	63.55 ± 22.99 ^a^	84.96 ± 33.29 ^b^	98.56 ± 40.47 ^a^	124.82 ± 57.65 ^c^
D × D	4.48 ± 1.97 ^b^	47.85 ± 20.08 ^b^	67.63 ± 25.13 ^a^	86.90 ± 27.21 ^b^	112.42 ± 35.49 ^ab^	142.66 ± 45.58 ^ab^
D × V	1.31 ± 0.88 ^a^	49.36 ± 20.85 ^c^	70.79 ± 28.32 ^b^	92.68 ± 33.76 ^c^	116.78 ± 50.53 ^b^	144.98 ± 59.51 ^a^
Heterosis (%)						
V × D growth	10.34	−6.22	−6.39	−6.66	1.59	2.98
D × V growth	−72.62	8.41	7.91	7.86	10.70	13.40
V × D survival	11.35	−0.28	−4.02	6.08	8.48	10.53
D × V survival	−19.47	−43.81	−38.71	−34.73	−19.01	−10.37

**Table 5 animals-10-01149-t005:** Specific growth rate and growth heterogeneity (mean ± SD) of the hybrid and purebred crosses of sterlet at 58, 175, 229, 325, 386, and 504 dph. Different superscripts within a row indicate significant differences.

Days Post Hatching	V × D (*n* = 240)	V × V (*n* = 240)	D × D (*n* = 240)	D × V (*n* = 240)
**Specific growth rate (% day^−1^)**				
58–175	1.77 ± 0.15 ^a^	1.82 ± 0.05 ^b^	2.02 ± 0.04 ^c^	3.10 ± 0.02 ^d^
176–229	0.69 ± 0.16 ^a^	0.72 ± 0.18 ^a^	0.64 ± 0.13 ^a^	0.67 ± 0.08 ^a^
230–325	0.28 ± 0.08 ^a^	0.30 ± 0.09 ^a^	0.26 ± 0.06 ^a^	0.28 ± 0.04 ^a^
326–386	0.49 ± 0.11 ^a^	0.23 ± 0.12 ^a^	0.43 ± 0.13 ^a^	0.37 ± 0.16 ^a^
387–504	0.23 ± 0.02 ^a^	0.21 ± 0.09 ^a^	0.20 ± 0.03 ^a^	0.18 ± 0.04 ^a^
**Growth heterogeneity**				
58–175	1.07 ± 0.10 ^a^	1.01 ± 0.02 ^b^	0.94 ± 0.05 ^c^	0.59 ± 0.04 ^d^
176–229	0.89 ± 0.09 ^a^	0.91 ± 0.02 ^a^	0.90 ± 0.08 ^a^	1.02 ± 0.01 ^a^
230–325	1.08 ± 0.07 ^a^	1.09 ± 0.17 ^a^	1.06 ± 0.17 ^a^	1.14 ± 0.16 ^a^
326–386	0.82 ± 0.07 ^a^	0.99 ± 0.19 ^a^	0.82 ± 0.12 ^a^	1.10 ± 0.19 ^b^
387–504	1.24 ± 0.10 ^a^	1.10 ± 0.10 ^ab^	1.02 ± 0.16 ^ab^	0.83 ± 0.10 ^b^

**Table 6 animals-10-01149-t006:** Summary statistics of the genetic variation among the analyzed purebred and hybrid crosses of the Volga and Danube sterlet.

Locus	H_o_	H_o_ SD	H_e_	H_e_ SD	N_A_	N_A_ SD
D × D	0.6553	0.0442	0.7536	0.0482	4.5	1.44
V × V	0.5919 *	0.0311	0.6250 *	0.0327	4.3 *	1.52
D × V	0.6962 *	0.0498	0.7589 *	0.0685	4.6 *	1.65
V × D	0.6398	0.0523	0.7462	0.0279	4.5	1.28

N_A_ = Mean no. of alleles; H_e_ = expected heterozygosity; H_o_ = observed heterozygosity. * indicates significant difference at *p* < 0.05.

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
