# Peer review of "Intraspecific Hybrids Versus Purebred: A Study of Hatchery-Reared Populations of Sterlet *Acipenser ruthenus"

_animals, 2020, doi:10.3390/ani10071149_

Round 1
Reviewer 1 Report
Sturgeons, including the sterlet Acipenser ruthenus, are among the most endangered fish species worldwide. Conservation programs might be forced to use non-native fish of different origin if native ones are lacking. In commercial aquaculture, intraspecific hybridization of genetically distinct populations might be performed to exploit heterosis effects on growth and survival. In both cases, information on the consequences of intraspecific hybridization is needed for an appropriate management but is missing so far for sterlet.
To close this knowledge gap authors first analyzed the genetic structure of hatchery-reared Danube and Volga populations of sterlet based on six microsatellite loci to ensure that the two were genetically distinct, and then investigated fitness-related traits (fertilization and hatching rates, growth, and survival) and genetic diversity of their purebred progenies and reciprocal hybrid crosses. They found 504 days after hatching the highest mean body weight in the Danube × Volga hybrid and the highest survival in Volga × Danube hybrid. In addition, the Danube × Volga hybrid also showed a higher genetic diversity than the purebred progenies. The Danube × Volga hybrid could therefore be recommended over purebreds for commercial aquaculture.
In my opinion the study is worth to be published. The methodological approach is basically appropriate and results are presented in logical order. Nevertheless, some revision is necessary (see detailed comments below) before final acceptance.
Material and methods
2.1. Fish populations
Better change heading to: Parental populations or parental brood stocks.
Please provide more background information, e.g. number of generations in captivity, effective population size which both have an influence on the within-population genetic variability estimated by the studied microsatellite loci.
2.4. Fertilization and hatching
Line 124: Delete “The breeding program is shown in Figure 1.” The breeding program and reference to figure 1 is already given in lines 114-115.
2.5. Rearing
Better extend heading to: Rearing of progeny groups
Was the dry feed also delivered ad libitum or in relation to fish biomass?
2.7. Molecular analysis
Better heading (i.e. more precise): Microsatellite marker analysis
What is the difference between lines 178-180 and 181-182? It looks like a duplication.
2.8. Statistical analysis
Instead of α use p for significance levels.
Lines 194-197 do not belong to this section. Either add a new section (2.9. Ethical approval) or move this part to the beginning of Material and methods after line 89.
Results
3.1. Population genetic analysis of Danube and Volga sterlet
GST and DA values are reported but their calculation is not described in Material and methods. Also, results of FCA analysis are presented in figure 2 but, again, this analysis is not described in Material and methods.
3.2. Performance comparison of purebreds and hybrids
Lines 225-226: “Although the fertilization rate was highest in the D × V hybrid, the hatching rate was significantly lower than in the Volga purebreds.” This is in contradiction to figure 3; I think you mean “…the hatching rate was significantly lower than in the Danube purebreds.”
3.3. Molecular analysis
Better heading (i.e. more precise): Microsatellite marker analysis
Line 257-258: “The observed … and expected … heterozygosity were high in the D × V hybrid (Table 6)”. I think, “highest” would be correct.
Figure 4: it would be better to use the same colors for purebreds as in figure 2, i.e. blue instead of grey for Danube sterlet.
Discussion
Lines 275-278: Information on the origin of parental brood stocks and the number of generations in captivity should be moved to Material and methods where it is missing.
Author Response
Dear Reviewer,
Herewith the copy of the addressed document for all your comments. Please find the attachment.
We hope you’ll be satisfied by our answers to your specific comments.
With Kind Regards,
Sahana

Reviewer 2 Report
This paper examines the impact of hybridisation of 2 populations of stugenon, one from the Volga and one from the Danube. This is done by undertaking a genetic cross, and monitoring several fitness measures. Overall I felt that this is well written and explained. Overall, no fitness disadvantages were seen in the sturgeon crossed from Volga and Danube populations, and there was some indication of increased fitness of the hybrids.
My only major comment is the details about rearing the offspring. How many replicate tanks were used. How are we sure that the observed effects are not due to the effects of the tank. This is alluded to in the discussion r.e. cannibalism, but should be made more explicit.
There were several points in the introduction that require clarification for readers not familiar with the system. E.g. once hatchery reared, are they then released into rivers, or are they kept in hatchery facilities till adulthood?
In the discussion, the limitations of the study need to be made cristal clear. This is important as the study could lead to management consequences with implications for populations. While it suggests that the hybridisation would benefit the population, are we 100% sure that this is the case. E.g. how much does growth indicate fitness, is high growth rate always a good thing for a fish. I think it is clear that higher hatch rate is beneficial. Another is the chemistry of the river water. If local adaptation is important, then changing the river water, levels of contamination could have an impact on the results. E.g. we know that several estuarine fish have adapted to PAHs PCB contamination in the USA. If this had occurred in any of these populations, this may not necessarily make a difference to these results, but if released back into the wild, this could have fitness implications, perhaps not for the hybrid, but for the F2 generation.
Minor points.
Abstract,
Units missing for the body mass of the fish.
Intro
I think it could do with the commercial value being placed higher up and what it is used for (for those that are unfamiliar to the subject). E.g. the global caviar industry is worth xxx, and this comes from x species of sturgeon.
Any data to back up “profound decline”, even if it is from historical documents etc?
Line 56: hence, may play a crucial role in stocking success [11,13,14]. – I am not convinced this has been demonstrated for fish i.e. an experiment showing different success for different levels of genetic variation. Undoubtedly genetic variation is important for adaptability though, just I’m not convinced the evidence is there yet. Perhaps be more explicit, just in case readers get the wrong message.
Line ~62: there are examples of genetic rescue involving other groups of animals (e.g. florida panther). Might be useful here to suggest that such approaches can work. (though whether it will work in this case is not known.
Important linked point – do sturgeon go back to where they hatched? How long do they spend at sea?
Another background point that readers may not be aware of is the size of these catchments, and the fact that that their catchments are enormous. This may deserve ½ a sentence.
Line 71 (approx). Some background info missing on hybrids. Are these fertile? Do the sturgeon used have the same number of chromosomes? “Heterosis is not considered detrimental to conservation efforts,”
Methods
Line 93: we need to know where the Volga and the Danube are and that they both flow into the Black sea. Useful to indicate somewhere (here or previously) any population-genetic work that has gone into demonstrating that these are different populations.
Line 98: How many microsatellite loci? – from a different study, or developed for this research?
Line 139: This is a clear diagram. Very useful. But it left me confused. In the methods it mentions 4 females were used, but here we have 12 females?
Line 144: How many replicate tanks were used for each cross. Tank can have a big influence (in my experience). Although I do appreciate it can be impractical to have large numbers of replicates in big experiments.
Line 183: What was the unit of replication? Was it 3 ‘tank’ replicates of each treatment? Being explicit here would help the reader.
Results
In general I would avoid the acronyms e.g. BW/SGR as it makes it hard to follow (unless there is a page limit reason for this).
Table 5 – perhaps as a graph, showing how survival changes over time for each group. This is quite hard to see in the table.
Line 263 – exactly how were the stats done for the Table 6. Was this an ANOVA, using the locus as a fixed effect? Here the number of significant figures is too high. I think that anything above 2 would be relatively meaningless.
Figure 4 – this is diffuclt to explain why V*D should be any different to D*V. Is it simply a function of having a limited number of offspring from each cross.
Discussion
I would suggest having the first paragraph as giving an overall summary of the main findings. This would help the skim-readers.
Is the advantage of using hybrids sufficiently large to warrant the risk of disrupting local adaptation?
For salmonids, work has shown that the underlying geology has a major impact on adaptation. E.g. for whether it is acidic or alkaline. This could have a much bigger influence than the actual river location. Is there any information about this for the broodstock?
Author Response
Dear Reviewer,
Thanks a lot for you valuable comments and suggestions. Herewith the copy of the addressed document for all your comments. Please find the attachment.
We hope you’ll be satisfied by our answers to your specific comments.
With Kind Regards,
Sahana

Reviewer 3 Report
The authors compared genetic and growth/survival differences among intraspecific hybrids and purebreds of hatchery-reared sterlet. Generally speaking, they collected lots of data to support their conclusions. The overwriting is Ok; however, extra editing and format improvement are required before acceptance for publication.
Minor issues:
Line 2: Replace the semicolon with a colon.
Lines 29-30: Are the two hybrids different? If it is true, please add more details to tell the exact difference. Please correct the same issues throughout the manuscript.
Figures 2-4: Please add numbers of examined fish.
In the discussion: Add subtitles to separate corresponding paragraphs.
Author Response

(The authors gave the same response as above.)
